# The Cardiorenal Effects of *Piper amalago* Are Mediated by the Nitric Oxide/Cyclic Guanosine Monophosphate Pathway and the Voltage-Dependent Potassium Channels

**DOI:** 10.3390/ph16111630

**Published:** 2023-11-20

**Authors:** Luciane M. Monteiro, Lislaine M. Klider, Aline A. M. Marques, Paulo V. Farago, Janaína Emiliano, Roosevelt I. C. Souza, Ariany C. dos Santos, Vera L. P. dos Santos, Mei Wang, Nadla S. Cassemiro, Denise B. Silva, Ikhlas A. Khan, Arquimedes Gasparotto Junior, Jane Manfron

**Affiliations:** 1Graduate Program in Pharmaceutical Sciences, State University of Ponta Grossa, Ponta Grossa 84030-900, PR, Brazil; lucianemendesmonteiro@gmail.com (L.M.M.); lis_klider19@hotmail.com (L.M.K.); pvfarago@gmail.com (P.V.F.); janemanfron@hotmail.com (J.M.); 2Laboratory of Cardiovascular Pharmacology (LaFaC), Faculty of Health Sciences, Federal University of Grande Dourados, Dourados 79825-070, MS, Brazil; alinemarques_nutri@hotmail.com (A.A.M.M.); rooseveltsouza@ufgd.edu.br (R.I.C.S.); arianysantos@ufgd.edu.br (A.C.d.S.); 3National Center for Natural Products Research, University of Mississippi, University, MS 38677, USA; ikhan@olemiss.edu; 4Laboratory of Natural Products and Mass Spectrometry (LaPNEM), Faculty of Pharmaceutical Sciences, Food and Nutrition (FACFAN), University of Mato Grosso do Sul (UFMS), Campo Grande 79080-190, MS, Brazil; janaemiliano@gmail.com (J.E.); nadlacass@hotmail.com (N.S.C.); denisebrentan@gmail.com (D.B.S.); 5School of Health, Environment, Sustainability and Humanity, Uninter International University Center, Curitiba 80020-110, PR, Brazil; verabiologa13@gmail.com; 6Natural Products Utilization Research Unit, Agricultural Research Service, United States Department of Agriculture, University of Mississippi, University, MS 38677, USA; mei.wang@usda.gov

**Keywords:** acute toxicology, cardiovascular effect, diuretic effect, jaborandi-manso, pariparoba, Piperaceae

## Abstract

*Piper amalago* L. is used in Brazilian traditional medicine to treat inflammation, chest pain, and anxiety. This study aimed to investigate the safety and the renal and cardiovascular effects of the volatile oil (VO) and the aqueous (AE) and hydroalcoholic (HE) extracts from *P. amalago*. The gas chromatography-mass spectrometry analyses identified 47 compounds in the VO, with β-cyclogermacrene, spathulenol, β-phellandrene, and α-pinene standing out. Among the 47 compounds also found in AE and HE by liquid chromatography-mass spectrometry, glycosylated flavones, organic acids, amino acids, and amides were highlighted. Some examples of these compounds are methoxy-methylenedioxy *cis*-cinnamoyl pyrrolidine, methoxy-methylenedioxy *trans*-cinnamoyl pyrrolidine, and cyclobutene-2,4-*bis*-(1,3-benzodioxol-5-methoxy-6-yl)-1,3-dicarboxapyrrolidide. The acute toxicity experiments were conducted on female rats (*n* = 5). The cardiorenal assays (*n* = 8) and evaluations of vasodilatory effects on the mesenteric vascular bed (*n* = 5) were conducted on male rats. In either extract or VO, there were no mortality or changes in relative weights or histopathological analysis of the organs. Urinary volume and renal electrolyte excretion were elevated significantly during repeated dose 7-day treatment with different preparations from *P. amalago*. None of the preparations induced hypotension or changes in cardiac electrical activity. Only HE promoted significant vasodilatory effects in rats’ isolated mesenteric vascular beds. These effects were completely abolished in the presence of L-NAME *plus* 4-aminopyridine. Therefore, *P. amalago* leaves are safe and present diuretic activity after acute and repeated dose administration over 7 days. Moreover, the HE induced significant vasodilator response in rats’ mesenteric vascular beds by NO/cGMP pathway and voltage-dependent K^+^ channels activation.

## 1. Introduction

*Piper amalago,* a member of the Piperaceae family, is commonly described as ‘pariparoba, jaborandi-manso, and pimenta-da-jamaica’ in Brazil. It is a native perennial tree that can grow up to a height of 7 m. This species, known for its aromatic properties, produces and stores volatile oil (VO) within its glandular trichomes and secretory cells [1,2,3]. In traditional Brazilian medicine, leaves of *P. amalago* are used as a diuretic for the treatment of urinary calculi illness and heart issues. They are also employed as a healing agent for burns, abscesses, boils, wounds, and insect bites. Furthermore, they are consumed to alleviate inflammation, chest pain, gastrointestinal problems, headaches, nosebleeds, and anxiety. Additionally, Puerto Ricans use masticated leaves of *P. amalago* to treat bleeding wounds, while the Huasteco-Maya tribes in Mexico use the leaves for their antipyretic, antiedemic, and anti-inflammatory properties [2,4].

Several biological activities have been previously reported for this species. These activities include antibacterial, anti-inflammatory, pain-relieving, anti-arthritis, anti-leishmanial, antioxidant, antilithiasic, anxiolytic, and insecticidal properties. In addition, it induces collagen production in mouse fibroblasts [2,4,5,6,7]. Moreover, various preparations made from *P. amalago* showed no signs of toxicity in rodents [8].

From a chemical perspective, this species is known for containing high amounts of amides, mostly pyrrolidines and isobutylamides. It also contains chalcones, alkaloids, triterpenes, condensed tannins, and flavonols [2,4]. The *P. amalago* VO also contains monoterpenes and sesquiterpenes [4]. In experiments involving rodents, the amides present in the ethanol extract from *P. amalago* have shown properties such as reducing hyperalgesia, alleviating pain, and treating arthritis [9].

No previous studies have investigated the cardiovascular effects of *P. amalago*, despite its widespread ethnopharmacological recognition. Regarding the use of this species as a diuretic, the objective of this study was to investigate the safety and the renal and cardiovascular effects of aqueous (AE) and hydroalcoholic (HE) extracts, as well as VO from *P. amalago* leaves, to shed light on the potential mechanism of action.

## 2. Results

### 2.1. Chemical Compounds of Aqueous and Hydroalcoholic Extracts

The AE and HE were analyzed using LC-DAD-MS. As a result, 47 compounds were identified by comparing the spectral data (UV, MS, and MS/MS) to the literature. The precise MS data were used to determine the molecular formula, with permissible errors of up to 7 ppm and 25 mSigma. A summary of all the spectral records can be found in Table 1.

The peaks 1–5 (Figure 1) showed the presence of deprotonated ions with mass-to-charge ratios of 165.0405, 179.0565, 191.0203, 178.0881, and 203.0825. These results indicated the molecular formulas C_5_H_10_O_6_, C_6_H_12_O_6_, C_10_H_13_NO_2_, and C_11_H_12_NO_2_. Therefore, these peaks were identified as pentonic acid (**1)**, hexose (**2**), citric acid (**3**), *O*-methyl phenylalanine (**4**), and tryptophan (**5**).

The compounds **6**, **8**–**9**, and **11**–**17** exhibited two absorption bands at approximately 270 and 330 nm, which are consistent with flavones [10]. Compound **17** displayed an additional absorption band at 322 nm, indicating the presence of the chromophore coumaroyl [11]. The loss of 90 and 120 u suggests the presence of C-hexosyl substituents, while the loss of 60 and 90 u indicates the presence of C-pentosyl. Moreover, the losses of 162, 132, 206, 176, and 146 u suggest the presence of *O*-hexosyl, *O*-pentosyl, *O*-sinapoyl, *O*-feruloyl, and *O*-coumaroyl substituents [10,11,12]. For instance, the observed fragment ions from *m*/*z* 801.2248 [M+H]^+^ (15) at *m*/*z* 433 [M+H-162-206]^+^, followed by the loss of water molecules, yielding ions at *m*/*z* 415, 397, and 379. The subsequent loss of water molecules is a common occurrence in C-glycosylated flavonoids [12], such as the product ions at *m*/*z* 379 [M+H-3xH_2_O]^+^ and 361 [M+H-4xH_2_O]^+^. All the aglycones of **6**, **8**–**9**, and **11**–**17** were identified as apigenin. Their spectral data align with the published literature [10,13]. *Piper* species have already been reported to contain C-glycosylated apigenins [4,14].

The peaks 19–28, 30–32, 34, 36–37, and 39–47 correspond to amides containing one or two nitrogen atoms (specifically, dimeric amides 21–22, 26–28, and 32). These types of amide alkaloids are commonly associated with the *Piper* genus [10,13] and classified as types A to E. Amides A to C have a group called 1,3-benzodioxol-5-yl on one side and a pyrrolidine, piperidine, or isobutylamine on the other side. On the contrary, amides D and E have an alkyl group on one side and an *N*-isobutylamine or piperidine on the other [15]. The presence of pyrrolidine, isobutylamine, and piperidine groups is indicated by fragment ions that result in losses of 71, 73, and 85 u, respectively [16,17]. Compounds **23** and **24** showed the fragment ions with *m*/*z* 229, caused by a loss of 71 u, suggesting the presence of the pyrrolidine group. Therefore, **23** and **24** were identified as *N*-[7-(3′,4′)-methylenedioxyphenyl-2,4, heptadienoyl]-pyrrolidine (nigrinodine) [18,19], which have already been described in *P. amalago* [9]. All the amide alkaloids identified from AE and HE are summarized in Table 1.

### 2.2. Chemical Compounds of Volatile Oil

The VO yield was 1.02%, while the density was 0.98 g/mL. GC-MS analysis was used to determine the chemical composition of VO, and all the compounds are summarized in Table 2. The identified compounds consisted of 17.02% hydrocarbon monoterpenes, 10.68% oxygenated monoterpenes, 36.17% hydrocarbon sesquiterpenes, and 34.04% oxygenated sesquiterpenes. The major compounds of the VO were bicyclogermacrene (11%), spathulenol (9.21%), *β*-phellandrene (8.12%), and α-pinene (6.17%).

### 2.3. Acute Toxicity

No noticeable alteration in the animal’s physical appearance or in the overall behavior was observed for a period of 14 days. There were no significant variances detected in the increase of body weight, water and food intake, or the relative weights of organs among the various experimental groups. The macroscopic examination did not reveal any indications of toxicity. The examination of organ samples through histopathological investigation exhibited no signs of fluid buildup, inflammation, or any other changes that may suggest injury (Figure 2).

### 2.4. Diuretic Effects

The results regarding the acute diuretic effects (8 and 24 h) of the AE, HE, and VO are presented in Table 3. The smallest dose (30 mg/kg) of AE and HE caused a significant increase in urinary volume 8 and 24 h after administration compared to the control. The higher dose (300 mg/kg) of AE and HE resulted in a significant increase in urine volume 8 h after taking it, compared to the control. As expected, HCTZ caused a significant increase in urine volume 8 h after administration and returned to levels similar to the control 24 h after treatment.

After a single oral dose, the AE at a dose of 30 mg/kg caused a decrease in the elimination of urea (8 h after treatment) and creatinine (8 h after treatment) from the kidneys. On the other hand, a higher dose (300 mg/kg) resulted in a significant increase in the elimination of calcium from the kidneys 24 h after administration, compared to the control group. The lowest dose (30 mg/kg) of HE significantly reduced the excretion of urea, creatinine, and potassium 8 h after administration. Similarly, the higher dose (300 mg/kg) of HE also caused a decrease in the elimination of urea, creatinine, and potassium 8 h after administration, compared to the negative control group. When looking at the group treated with VO, the lowest dose (30 mg/kg) caused a significant increase in the excretion of urea (24 h after treatment), creatinine (24 h after treatment), magnesium (8 h after treatment), sodium (24 h after treatment), and chloride (24 h after treatment) compared to the control group (Table 4).

The effects of administering AE, HE, and VO orally for 7 days (on days 3 and 7) and their impact on urinary volume and electrolyte excretion (24 h after treatment) can be found in Table 5 and Table 6, respectively. Administering the lowest dose (30 mg/kg) of AE for 7 days resulted in a significant increase in urinary excretion on the 7th day. The VO caused a significant increase in urinary volume for both doses (30 and 300 mg/kg) on the 7th day of treatment. HCTZ also caused a significant increase in urinary excretion 24 h after administration on the 7th day.

The AE, administered at a dose of 30 mg/kg, caused a significant increase in the elimination of urea, chloride, potassium, and sodium from the kidneys on the 7th day. At the higher dose (300 mg/kg), AE caused an increase in the excretion of calcium on the 3rd day compared to the negative control. In the group treated with HE at a dose of 30 mg/kg, there was an observed increase in the renal excretion of chloride, potassium, and sodium on the 7th day. The highest dose of HE, 300 mg/kg, significantly increased the elimination of potassium from the kidneys on the 3rd day of treatment. Oral administration of VO at a dose of 30 mg/kg resulted in a significant increase in the excretion of creatinine, urea, and potassium on the 3rd day. However, VO at the same dose significantly reduced calcium excretion on the 7th day after the treatments. There were no changes observed in density and pH values with any of the treatments. At the end of the treatments, no significant effects were observed on serum biochemical parameters (Table 7).

### 2.5. Electrocardiography

The electrocardiographic waves and intervals did not exhibit any significant changes among the various experimental groups. The segments PR, QRS, QT, and QTc, as well as the waves P, Q, R, and S, remained unaltered when treated with AE, HE, VO, HCTZ, or the vehicle (negative control) (Table 8).

### 2.6. Effects on Blood Pressure and Heart Rate

After seven days of treatment, the animals receiving AE, HE, VO, or HCTZ did not show any significant reduction in SBP, DBP, MAP, and HR values compared to the negative control rats (as shown in Figure 3A–D).

### 2.7. Vasodilator Response in MVBs

The administration of different doses (0.3, 1, and 3 mg) of HE in MVBs caused a vasodilator effect that increased with higher doses (Figure 4A). The data revealed that the vasorelaxant effect of HE (at a dose of 3 mg) had the same impact as the administration of 1 nmol of Ach. Treatment with sodium deoxycholate reduced the effects of ACh by 97 ± 2% in MVBs. Furthermore, the effect of AE doses was significantly reduced in preparations that were treated with sodium deoxycholate, as shown in Figure 4B. Pre-treatment with L-NAME in preparations with a functioning endothelium also reduced the vasodilating effect of HE (Figure 5A). In preparations with a functional endothelium that was perfused with indomethacin, TEA, or GLB, the vasorelaxant effects of HE remained unchanged (Figure 5B–D). The use of 4-AP during perfusion significantly decreased the effects of HE (Figure 5E). Furthermore, in preparations perfused with both 4-AP and L-NAME, the vascular effects of HE were completely abolished (Figure 5F). Lastly, the administration of AE or VO did not induce any vasodilator response in the MVBs.

## 3. Discussion

Brazil, with its vast biodiversity, could potentially contribute to the development of new medicines derived from medicinal plants [3]. The National Policy of Integrative and Complementary Practices in Brazil’s unified health system aims to incorporate both traditional and alternative complementary medicine into all public health services across the country. The World Health Organization acknowledges that a significant portion of the population in developing nations relies on folk medicine for primary healthcare. However, modern Western medicine is extensively utilized in most of the world, which unfortunately undermines traditional medicine [21].

The active compounds found in plant species are chemical components that have the potential to induce a biological reaction or influence the bodies of humans and animals. These compounds are typically divided into various groups based on their chemical properties and biological impacts. Factors like environmental conditions, genetic markers, variations in plant physiology, phenological factors, and the extraction method employed can affect these plant compounds [22]. In our study, we initially analyzed the secondary metabolites found in AE and HE. We identified a wide range of compounds, including glycosylated flavones, organic acids, amino acids, and amides such as methoxy-methylenedioxy-*cis*-cinnamoyl pyrrolidine, methoxy-methylenedioxy *trans*-cinnamoyl pyrrolidine, and cyclobutene-2,4-*bis*-(1,3-benzodioxol-5-methoxy-6-yl)-1,3-dicarboxy pyrrolidide. Different classes of chemical compounds, including alkaloids, amides, benzoic acid derivatives, chalcones, chromenes, flavones, flavanones, kavalactones, lignans, neolignans, terpenes, steroids, and fatty acids were found in *Piper* species [2,4,22]. This demonstrated the diverse biosynthetic pathways present in this genus.

Regarding the level of VO, monoterpenes and sesquiterpenes are often identified in *Piper* species [2,4,22,23]. However, sesquiterpenes are more abundant in the genus, as seen in *P. arboreum* Aubl., *P. dilatatum* Rich., *P. hispidum* Sw., *P. aduncum* L., *P. amalago* L., *P. cernuum* Vell., and *P. regnelii* (Miq.) C.DC. [4,22]. In our study, 97.91% of the volatile chemical compounds were identified. These compounds consisted of 17.02% monoterpene hydrocarbons, 10.64% oxygenated monoterpenes, 36.17% sesquiterpene hydrocarbons, and 34.04% oxygenated sesquiterpenes. Other studies on *P. amalago* leaves have found different compounds, including α-amorphene (25.7%), cubenol (6.2%), and methyl geranate (7.8%) [4,22]. Volatile organic compounds have also been extracted from other organs of *P. amalago*, displaying different chemical profiles. VO extracted from *P. amalago* inflorescences showed α-muurolol (5.0%), *p*-cymene (9.3%), limonene (10.5%), silfiperfol-6-eno (13.5%), and *allo*-aromadendrene (18.5%) as the major compounds [4]. To explore the diversity of chemical profiles in VO from *Piper* species, Thin et al., 2018 [23] proposed six groups based on the dominant classes of chemical compounds, including monoterpenes, sesquiterpenes, monoterpenes plus sesquiterpenes, phenylpropanoids, benzenoids, and non-terpenoid compounds. Accordingly, *P. amalago* VO belongs to the sesquiterpene class based on its chemical profile.

After analyzing the different preparations for their phytochemical properties, we evaluated the safety profile of AE, HE, and VO when given to female rats in a single dose. We conducted acute toxicity studies to determine the potential for lethality and to identify any short-term adverse effects. Based on our study, we found that AE, HE, and VO were safe for acute administration, as the median lethal dose was greater than 2000 g/kg. These findings support the results of a study conducted by Lopes et al. (2012) [8], which also showed the safety of *P. amalago* extracts when given in a single dose.

After demonstrating apparent safety in the acute toxicity study, a series of efficacy studies were conducted to evaluate various renal and cardiovascular parameters. Initially, noticeable diuretic effects were observed following the administration of AE, HE, and VO. It was noted that the acute diuretic effect of AE and HE does not depend on dosage, as the lower dose (30 mg/kg) of both substances significantly increased urine volume 8 and 24 h after oral administration, while the higher dose (300 mg/kg) only resulted in a significant increase in urine volume 8 h after administration. The diuretic activity exhibited by the lowest dose of the extract (30 mg/kg) may be linked to the phytocomplex, which acts independently on the active site. The phytocomplex refers to all components derived from primary or secondary metabolism, which are collectively responsible for the biological effects of a particular medicinal plant species or its derivatives [24]. Additionally, it is worth noting that AE and HE induced significant solute retention upon acute administration, which is likely due to fluid balance, which relies on fluid intake and urinary excretion [25]. Furthermore, it was observed that urine volume and solute excretion increased significantly, independent of dosage, on days 3 and 7. This is a common effect of diuretics, as they initially cause dehydration, leading to compensatory solute retention to maintain serum osmolarity. Consequently, the effect recurs when a new dose is administered and hydroelectrolytic balance is restored [26]. Novaes et al., 2014 [27] demonstrated that acute oral administration of the ethanolic extract obtained from *P. amalago* increased urinary volume and excretion of sodium and potassium. Additionally, it exhibited antilithiasic activity by reducing the size of calcium oxalate crystals.

Based on the results obtained from the diuresis tests, we decided to assess the effects of AE, HE, and VO on blood pressure. This choice was influenced by the fact that many traditional antihypertensive medications, such as HCTZ, have diuretic properties [26]. Although Iwamoto et al. (2014) have demonstrated that a compound isolated from *P. amalago* (*N*-[7-(3′,4′-methylenedioxyphenyl)-2(Z),4(Z)-heptadienoyl] pyrrolidine) exhibits antihypertensive and vasodilatory effects in spontaneously hypertensive rats [5], none of the tested extracts were able to lower blood pressure in the animals involved in our study. Interestingly, even HCTZ did not show a hypotensive response after seven days of oral treatment. Most traditional antihypertensive drugs usually require around four weeks to significantly reduce blood pressure levels, and this is due to various counter-regulatory mechanisms within the body, including the renin-angiotensin-aldosterone system. Additionally, it is worth noting that normotensive patients do not always experience significant blood pressure reduction compared to hypertensive individuals. This divergence can be attributed to the fact that hypertensive individuals have more active pressure molecular pathways. Consequently, using rat models of hypertension, such as spontaneously hypertensive animals, in studies provides an ideal environment for evaluating different molecular aspects of antihypertensive drugs and offers stronger foundational evidence for future clinical trials [26].

Although the extracts do not have hypotensive properties, there is evidence indicating that the compound *N*-[7-(3′,4′-methylenedioxyphenyl)-2(*Z*),4(*Z*)-heptadienoyl] pyrrolidine, which is isolated from *P. amalago*, may act as a vasodilator [5]. Based on this premise, the vasodilatory effects of the extracts on MVBs were investigated. Only the HE was able to induce a significant vasodilator response in MVBs. These effects seemed to be dependent on the release of endothelial nitric oxide and the opening of voltage-dependent potassium channels. Potassium channels play a critical role in regulating the membrane excitability of various cells, including vascular smooth muscle cells found in the MVBs. Consequently, potassium efflux increases when the potassium channels are opened, leading to membrane hyperpolarization and vasodilation [28]. Although the mechanisms behind the diuretic activity of these extracts remain unknown, based on the data collected from MVBs, it is possible to theorize that the HE may cause vasodilation that extends to the afferent renal arteriole, resulting in an increase in blood hydrostatic pressure and glomerular filtration rate, thus inducing a diuretic response.

There were two main limitations in this study. First, we were unable to determine if any of the secondary metabolites we identified could be solely responsible for the observed pharmacological effects. While it is an interesting hypothesis, we believe that the biological effects of *P. amalago* are more likely the result of a combined and coordinated action of multiple secondary metabolites. Second, we did not investigate the effects of the *P*. *amalago* extracts in an animal model of hypertension for a longer treatment period (4 or 8 weeks). By conducting further studies, we can gain more clarity on these matters and enhance our understanding from a comprehensive preclinical standpoint.

## 4. Material and Methods

### 4.1. Chemicals

Ketamine hydrochloride and xylazine were obtained from Syntec Pharma in Barueri, São Paulo, Brazil. Heparin was acquired from Hipolabor Pharmaceutical Manufacturing in Belo Horizonte, Minas Gerais, Brazil. The chemical compounds ethylene-diaminetetra acetic acid, calcium chloride, acetylcholine chloride, dextrose, *Nω*-Nitro-*L*-arginine methyl ester, phenylephrine, hydrochlorothiazide, glibenclamide, indomethacin, potassium chloride, potassium dihydrogen phosphate, magnesium sulfate, sodium chloride, sodium bicarbonate, and 4-aminopyridine and the volatile components α-humulene, α-pinene, α-terpineol, camphene, *β*-caryophyllene, *β*-eudesmol, *β*-myrcene, *β*-phellandrene, *β*-pinene, caryophyllene oxide, globulol, limonene, *p*-cymene, spathulenol, terpinen-4-ol, valencene, and viridiflorol were purchased from Sigma-Aldrich in St. Louis, MO, USA.

### 4.2. Botanical Material and Extract Preparation

*P. amalago* leaves were collected in Curitiba, PR, Brazil, in February 2019. The coordinates for the collection site are 25°23′53″ S and 49°20′36″ W, with an elevation of 980 m. The authorization code AC1A4A9 was obtained from CGEN/SISGEN to legally access the plant material. A taxonomist identified the plant material, and a sample was deposited at the Municipal Botanical Museum with the registration number MBM 71947. To extract VO, 200 g of dried leaves was subjected to hydrodistillation using a Clevenger apparatus for a duration of 4 h. Na_2_SO_4_ was used to dry the extracted VO at the end of each distillation process. The VO was stored at −4 °C in vials with Teflon caps. Additionally, *P. amalago* leaves were oven-dried at a temperature of 30 °C for three days and then fragmented to obtain the AE. To prepare the AE, 100 g of the sample was infused in 1 L of water heated to 88–90 °C. The infusion was allowed to cool down to room temperature (20–23 °C) over approximately 5 h. The remaining extract was concentrated to a volume of 200 mL using a rotary evaporator. To precipitate proteins and polysaccharides, 600 mL of ethanol was added to the extract. The precipitate was then freeze-dried and stored in a freezer at −20 °C. The HE was obtained through maceration. The amount of 100 g of dry and powdered leaves was soaked in a mixture of 70% ethanol and 30% water for 7 days. The solution was filtered, concentrated, and freeze-dried following the method described by Pereira et al. (2015) [29].

### 4.3. Chemical Profile of Extracts and VO from P. amalago

#### 4.3.1. Liquid Chromatography Coupled to a Diode Array Detector and Mass Spectrometry (LC-DAD-MS) Analysis

Shimadzu Prominence coupled with a diode array detector (DAD) and a mass spectrometer equipped with electrospray ionization, quadrupole analyzer, and time-of-flight in a MicrOTOF-Q III (Bruker Daltonics, Billerica, MA, USA). A C18 chromatographic column with dimensions of 150 × 2.1 mm and a particle size of 2.6 µm (Phenomenex, Torrance, CA, USA) was used, along with a flow rate of 0.3 mL/min, an oven temperature of 50 °C, and a mobile phase consisting of ultrapure water (A) and acetonitrile (B) with 0.1% (*v*/*v*) formic acid in both solvents A and B. The gradient elution profile was as follows: 0–2 min—3% B; 2–25 min—3–25% B; 25–40 min—25–80% B; and 40–43 min—80% B. Prior to injection into the chromatographic system, the samples (2 µL at 2 mg/mL) were filtered through a 0.22 µm PTFE syringe filter. Both negative and positive ion modes were applied to the samples, with a capillary voltage of 4.500 V. Nitrogen gas was used as the dry gas (9 L/min), nebulizer (4.0 Bar), and collision gas.

#### 4.3.2. Gas Chromatography Coupled to Mass Spectrometry (GC-MS) Analysis

The VO samples were analyzed using an Agilent 7890A gas chromatography (GC) instrument, which was connected to an Agilent 5975C mass spectrometer (Agilent, Santa Clara, CA, USA). The capillary column (60 m × 0.25 mm inner diameter × 0.25 µm film thickness) used was a J&W HP-5MS. Helium was utilized as the carrier gas at a flow rate of 1 mL/min. Each sample was analyzed under the following conditions: initially at 50 °C for 2 min, and then heated gradually at rates of 2 °C/min (until reaching 170 °C), 1 °C/min (maintained at 170 °C), and 8 °C/min (until reaching 250 °C). The inlet temperature was set to 280 °C with a split ratio of 25:1. The mass spectrometer operated with an electron energy of 70 eV. Throughout the experiment, the temperatures of the ion source, quadrupole, and transfer line were maintained at 230 °C, 150 °C, and 280 °C, respectively. The mass spectra were recorded from *m*/*z* 35 to 500 after a 5 min delay in a solvent. Compound identification was conducted by comparing the mass spectra to databases (Wiley and NIST), and additional identification involved comparing relative retention rates with Adams (2017) [20] and utilizing benchmarks obtained from commercial sources or those internally isolated.

### 4.4. Pharmacological and Toxicological Studies

#### 4.4.1. Animals

We obtained Wistar rats (both males and females) that were 134 days old from the central vivarium of UFGD. These rats were kept in polypropylene cages measuring 49 cm × 34 cm × 16 cm, with sterilized wood shavings, and under a light/dark cycle at a temperature of 22 ± 3 °C. The rats had free access to food and water. The number of rats per group varied between 5 and 8 individuals depending on the experimental protocol. We chose rats as our experimental model based on the studies conducted by Gasparotto Junior et al., 2009 [30]. All experimental procedures involving the animals had been approved and licensed by CEUA, UFGD (protocol n° 28/2019). All procedures were conducted in accordance with the Brazilian Guidelines for the use of animals.

#### 4.4.2. Safety Evaluation

##### Acute Toxicity

The evaluation of acute oral toxicity was conducted on female rats according to protocol no. 425 by the Organization for Economic Co-operation and Development (OECD, 2008) [31]. In this experiment, 20 rats (*n* = 5) were divided into four groups. The substances AE, HE, and VO were given orally (by gavage) in a single dose of 2000 mg/kg. The negative control group received only filtered water (vehicle) in the same volume used to dilute the extracts. The animals were closely observed for 14 days, with their water and food intake, body weight gain, and signs of toxicity being measured daily. The animal’s behavior was observed using the Hippocratic screening method developed by Malone and Robichaud (1962) [32]. On the 15th day, an overdose of isoflurane was used to euthanize the animals. The kidney, heart, liver, lung, and spleen were then removed, weighed, and examined macroscopically. These organs were later sent for histopathological analysis.

#### 4.4.3. Ethnopharmacological Investigations

##### Diuretic Activity

This experiment followed the methods described by Gasparotto Junior et al. 2009 [30]. Male rats were randomly assigned to eight groups (*n* = 8): (a) AE (30 and 300 mg/kg); (b) HE (30 and 300 mg/kg); (c) VO (30 and 300 mg/kg); (d) HCTZ (hydrochlorothiazide; 25 mg/kg); and a negative control (vehicle; 0.2 mL/100 g). The doses of the extracts were based on the most used preparation in Brazil. In this preparation, 200 mL of boiling water is poured directly onto a crushed amount of the plant equivalent to a closed hand. A closed handful of *P. amalago* leaves weighs approximately 2.5 g. If we divide 2.5 g of *P. amalago* dry leaves by the weight of an adult human (70 kg), we obtain approximately 30 mg/kg. Therefore, we used a dose of 30 mg/kg and a ten-fold higher dose (300 mg/kg) as a safety measure. The animals were treated at 8:00 a.m. for seven days. Prior to the first treatment, all rats received a 0.9% sodium chloride solution (5 mL/100 g) to ensure consistent salt and water levels in their bodies. The rats were then housed in metabolic cages, and urine was collected after 1, 2, 4, 6, 8, and 24 h for seven days using a graduated cylinder. The total volume, density, pH, and urinary and serum levels of sodium, potassium, calcium, magnesium, urea, and creatinine were measured. The urinary levels of chloride were also determined.

##### Electrocardiography

After completing the diuresis experiment, all animals were anesthetized using an intramuscular route with 100 mg/kg of ketamine and 20 mg/kg of xylazine. To perform the ECG, electrodes were placed on the four limbs of the animal. The ECG waves were detected for an additional 5 min after an initial 5 min acclimatization period. The data were recorded using a digital ECG recorder (WinCardio, Micromed, Brasília, Brazil) and included the P, Q, R, and S waves, as well as the PR, QRS, QT, and QTc intervals.

##### Blood Pressure (BP) and Heart Rate (HR) Evaluation

For this procedure, we used the same animals that underwent electrocardiography. Initially, while still under anesthesia, we administered 30 IU of heparin subcutaneously. Next, we inserted a catheter into the left common carotid artery and connected it to a pressure recording system (PowerLab and LabChart setup, ADInstruments, Bella Vista, NSW, Australia). After waiting 15 min for the blood pressure to stabilize, we recorded the levels of diastolic blood pressure (DBP), systolic blood pressure (SBP), mean arterial pressure (MAP), and heart rate (HR) for 5 min [33]. At the conclusion of this procedure and after collecting a 5 mL sample of arterial blood, all animals were euthanized with an overdose of isoflurane.

##### Evaluations of Vasodilator Response in the Mesenteric Vascular Beds (MVBs)

MVBs (mesenteric vascular beds) from normotensive rats (*n* = 5) that had not undergone any previous treatment were quickly isolated and prepared for perfusion according to the methods described by Kawasaki et al. in 1988 [16] and 1991 [17]. The superior mesenteric artery was cannulated and gently washed with a physiological salt solution (PSS), which contained the following composition in millimoles per liter: 119 mM NaCl, 4.7 mM KCl, 2.4 mM CaCl_2_, 1.2 mM MgSO_4_, 25 mM NaHCO_3_, 1.2 mM KH_2_PO_4_, 11.1 mM dextrose, and 0.03 mM EDTA. The isolated MVBs were then placed in an organ bath, maintained at 37 °C, and continuously perfused with PSS solution at a constant 4 mL/min flow rate. The perfusion pressure was recorded using a pressure transducer connected to a computerized polygraph system (PowerLab and LabChart setup, ADInstruments, Bella Vista, NSW, Australia). After allowing 30 min for the preparations to stabilize, a PSS solution containing phenylephrine (Phe) at a concentration of 3 µM was continuously perfused. Once the tone of the preparations stabilized, the presence of functional endothelium was confirmed by administering acetylcholine (ACh) at a concentration of 1 nmol. Then, different doses of AE, HE, and VO at concentrations of 0.0003, 0.001, 0.003, 0.01, 0.03, 0.1, 0.3, 1, and 3 mg were injected in bolus into the perfusion system, and the resulting reduction in pressure was recorded. After another 30 min equilibrium period, other preparations were perfused with PSS containing 3 µM of Phe in combination with various agents, either used alone or in combination. These agents included L-NAME at a concentration of 100 µM (a non-selective inhibitor of nitric oxide synthase), indomethacin at a concentration of 1 µM (a non-selective inhibitor of cyclooxygenase), tetraethylammonium at a concentration of 1 mM (TEA, a non-selective blocker of calcium-sensitive potassium channels), 4-aminopyridine at a concentration of 10 µM (4-AP, a blocker of voltage-dependent potassium channels), and glibenclamide at a concentration of 10 µM (GLB, a selective blocker of ATP-sensitive potassium channels, specifically Kir6.1). To denude the endothelium of the MVB, some preparations were perfused with PSS containing sodium deoxycholate at a concentration of 1.8 mg/mL for 30 s via a parallel perfusion line connected to the central cannula. After the infusion of sodium deoxycholate, the system was perfused with regular PSS for an additional 40 min for stabilization. The effectiveness of sodium deoxycholate in removing the endothelium was confirmed by the absence of a reduction in perfusion pressure following the bolus injection of ACh at a concentration of 1 nmol. After 15 min of continuous perfusion, the extracts (AE, HE, and VO) were again injected into the system. The ability of the tested extracts to reduce perfusion pressure in the presence and absence of different inhibitors was evaluated.

### 4.5. Statistical Analysis

The Shapiro–Wilk and Levene’s tests were used to assess the normal distribution and homogeneity in variance, respectively. Statistical analyses were conducted using one-way ANOVA, followed by Dunnett’s test. The significance level chosen was 95% (*p* < 0.05). The analyses and graphics were conducted using GraphPad Prism version 9.5.0 for macOS.

## 5. Conclusions

Based on our data, the compounds AE, HE, and VO derived from *P. amalago* leaves are considered safe and have urinary effects when given to rats in both single and multiple doses over a 7-day period. Additionally, the HE showed a significant vasodilator effect in rats’ mesenteric vascular beds. This effect appears to be due to the activation of the NO/cGMP pathway and the voltage-dependent potassium channels.

## Figures and Tables

**Figure 1 pharmaceuticals-16-01630-f001:**
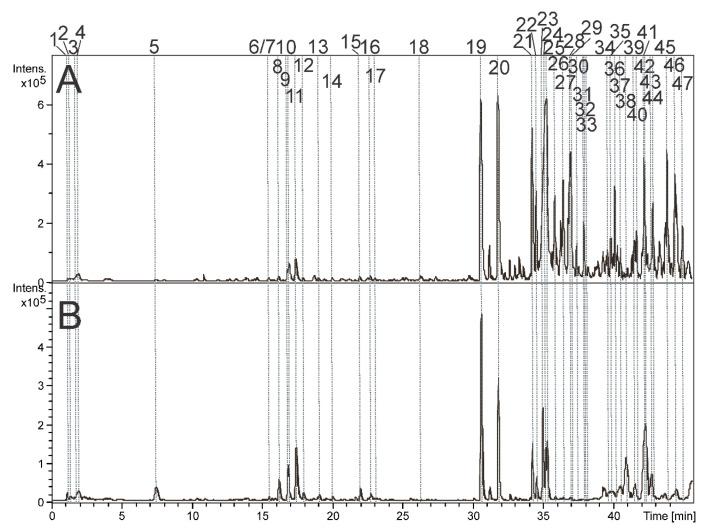
Base peak chromatograms were obtained in positive ion mode from the aqueous (**A**) and hydroalcoholic (**B**) extracts of *P*. *amalago* leaves. Details of peak numbers are described in the text.

**Figure 2 pharmaceuticals-16-01630-f002:**
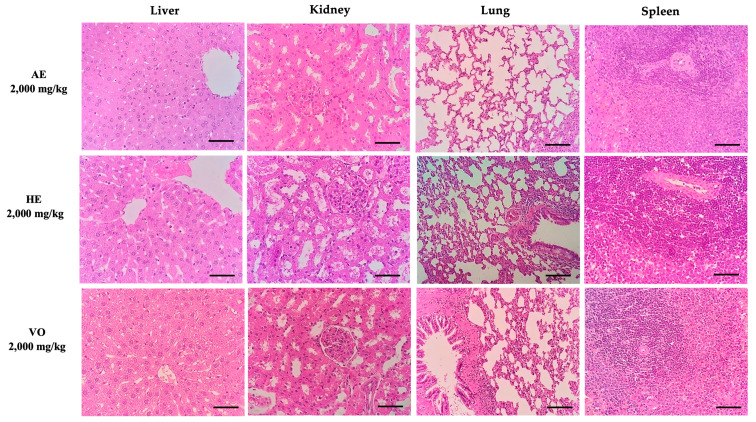
Histopathological evaluation of organs from female rats treated with a single oral dose (2000 mg/kg) of AE, HE, and VO in the acute toxicity test. H&E stain (40×). Scale bars = 100 μm. AE: aqueous extract; HE: hydroalcoholic extract; VO: volatile oil.

**Figure 3 pharmaceuticals-16-01630-f003:**
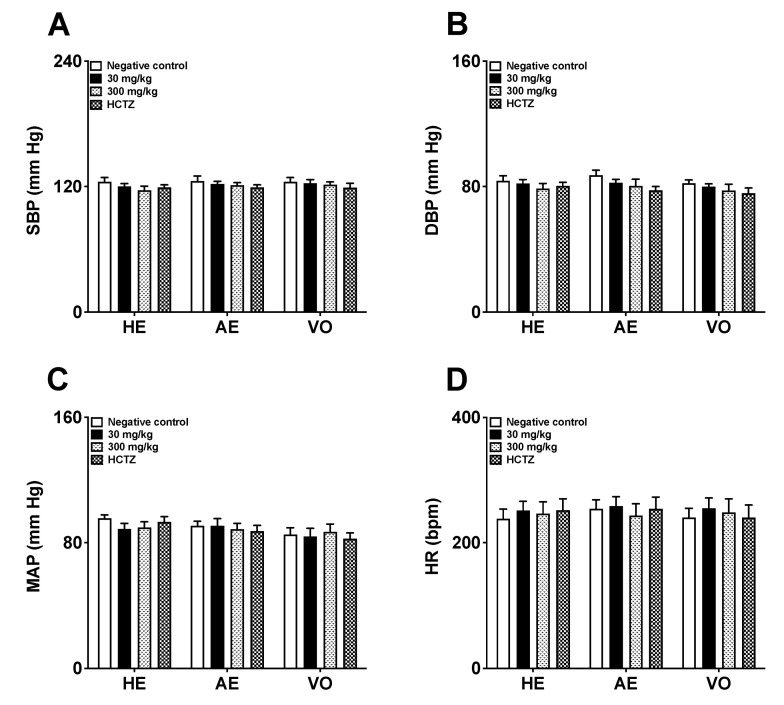
Effects of repeated 7-day oral dose administration of AE, HE, and VO obtained from *Piper amalago* leaves on blood pressure and heart rate ((**A**): SBP—systolic blood pressure; (**B**): DBP—diastolic blood pressure; (**C**): MAP—mean arterial pressure; (**D**): HR—heart rate). Statistical analysis was performed using one-way ANOVA followed by Dunnett’s test. Values are expressed as mean ± E.P.M. (*n* = 6). AE: aqueous extract; HCTZ: hydrochlorothiazide; HE: hydroalcoholic extract; VO: volatile oil.

**Figure 4 pharmaceuticals-16-01630-f004:**
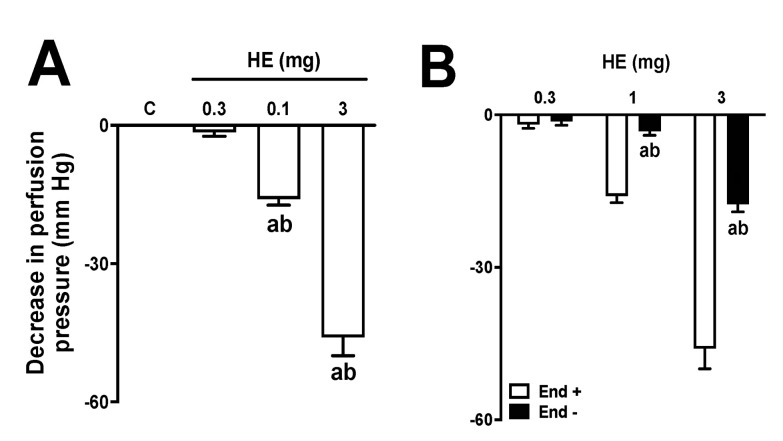
Effects of hydroalcoholic extract (HE) on the vascular tone of mesenteric vascular beds (MVBs) in the presence and absence of endothelium. MVBs were perfused with PSS containing Ach (3 μM) in the intact (**A**) or denuded endothelium (**B**), and the vasorelaxant effect of HE was evaluated. The results show the mean ± standard error of the mean of 4 or 5 preparations. In graph A, the letter a indicates *p* < 0.05 compared to the control group (vehicle); b indicates *p* < 0.05 compared to the previous dose. In graph B, the letter a indicates *p* < 0.05 compared to the effects of HE on the intact endothelium; b indicates *p* < 0.05 compared to the previous dose. Statistical analysis was performed using one-way ANOVA followed by Dunnett’s test. End − and End +: denuded and intact endothelium, respectively.

**Figure 5 pharmaceuticals-16-01630-f005:**
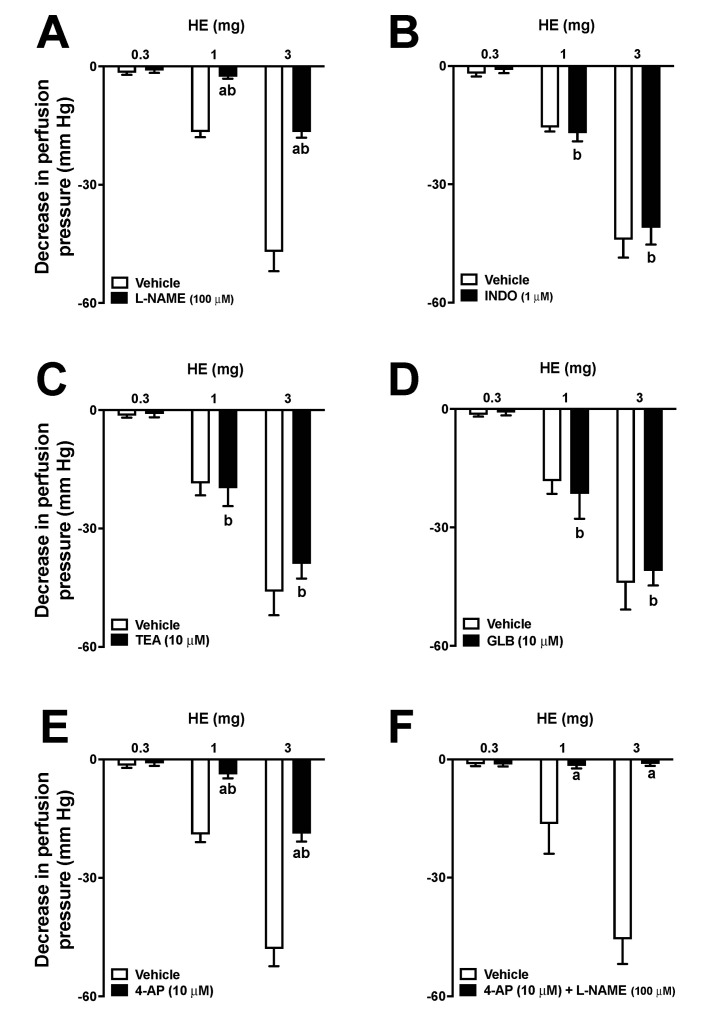
Effects of hydroalcoholic extract (HE) on the vascular tone of mesenteric vascular beds (MVBs) on the presence and absence of L-NAME (100 μM) (**A**), indomethacin (1 μM) (**B**), TEA (10 μM) (**C**), glibenclamide (10 μM) (**D**), 4-AP (10 μM) (**E**), and 4-AP (10 μM) + L-NAME (100 μM) (**F**). MVBs were perfused with PSS containing Phe (3 μM), and the vasorelaxant effect of HE was evaluated in the absence and presence of inhibitors. Results show the mean ± standard error of the mean of 4 or 5 preparations. The letter a indicates *p* < 0.05 compared to the respective control group (without inhibitor). The letter b indicates *p* < 0.05 compared to the previous dose (in the presence of an inhibitor). Statistical analysis was performed using one-way ANOVA followed by Dunnett’s test.

**Table 1 pharmaceuticals-16-01630-t001:** Compounds assigned from the aqueous (AE) and hydroalcoholic (HE) extracts of *Piper amalago* leaves by LC-DAD-MS.

Peak	RT (min)	UV (λmax)	MF	Negative Mode (*m*/*z*)		Positive Mode (*m/z*)		Compound	AE	HE
				MS [M-H]^−^	MS/MS	MS [M+H]^+^	MS/MS			
1	1.2	-	C_5_H_10_O_6_	165.0405	-	-	-	Pentonic acid	X	X
2	1.4	-	C_6_H_12_O_6_	179.0565	-	203.0537 ^Na^	-	Hexose	X	
3	1.6	-	C_6_H_8_O_7_	191.0203	-	-	-	Citric acid	X	X
4	1.9	277	C_10_H_13_NO_2_	178.0881	-	180.1024	163	*O*-Methyl-phenylalanine	X	X
5	7.4	270, 280, 287	C_11_H_12_N_2_O_2_	203.0825	-	205.0978	188, 170, 159, 146	Tryptophan	X	X
6	15.4	269, 337	C_27_H_30_O_15_	593.1527	-	595.1658	457, 409, 403, 391, 379, 355, 337, 325, 307, 295	Apigenin di-C-hexoside	X	X
7	15.4	283	C_14_H_19_NO_2_	232.1349	-	234.1486	217, 206, 188, 175, 161	Unknown		X
8	16.1	270, 335	C_27_H_30_O_15_	593.1515	503, 473, 413, 383, 353	595.1661	457, 439, 421, 409, 379, 355, 337, 325, 295	Apigenin di-C-hexoside	X	X
9	16.8	270, 335	C_27_H_30_O_15_	593.1529	503, 473, 431, 413, 383, 353, 341, 311	595.1681	415, 397, 379, 337, 313, 283	Apigenin C-hexosyl *O*-hexoside	X	X
10	16.9	283	C_14_H_21_NO_2_	234.1500	-	236.1654	219, 201, 177, 163, 149, 145	Unknown		X
11	17.3	270, 337	C_27_H_30_O_15_	593.1527	503, 473, 431, 413, 353, 341, 311, 297,	595,1657	415, 397, 379, 367, 361, 349, 337, 313, 283	Apigenin C-hexosyl *O*-hexoside	X	X
12	17.9	271, 336	C_26_H_28_O_14_	563.1411	413	565.1558	529, 511, 493, 475, 451, 445, 433, 427, 421, 409, 403, 397, 391, 379, 355, 349, 337, 325, 295	Apigenin *O*-pentosyl di-C-hexoside	X	X
13	19	270, 333	C_26_H_28_O_14_	563.1415	-	565.1560	385, 367, 349, 337, 325, 321, 313, 283	Apigenin *O*-hexosyl C-pentosyl C-hexoside	X	X
14	19.9	268, 337	C_21_H_20_O_10_	431.1001	-	433.1130	379, 361, 341, 337, 333, 323, 313, 309, 283	Apigenin C-hexoside	X	X
15	21.9	270, 337	C_38_H_40_O_19_	799.2107	-	801.2248	433, 415, 397, 379, 369, 337, 313, 283, 207, 175	Apigenin *O*- sinapoyl- hexosyl C-hexoside	X	X
16	22.6	272, 334	C_37_H_38_O_18_	769.1997	-	771.2157	475, 433, 415, 397, 379, 367, 361, 349, 337, 313, 283, 177	Apigenin *O*-feruloyl-hexosyl C-hexoside	X	X
17	23.0	271, 322	C_36_H_36_O_17_	739.1897	-	741.2017	415, 397, 379, 367, 361, 349, 337, 313, 283, 147	Apigenin *O*-coumaroyl-hexosyl C-hexoside	X	X
18	26.2	286, 319	C_18_H_19_NO_4_	312.1254		314.1392	177, 149	*N*-trans-feruloyl tyramine		X
19	30.6	271, 329	C_15_H_17_NO_4_	-	-	276.1233	205, 190, 175, 162	Methoxy-methylenedioxy *cis*-cinnamoyl pyrrolidine	X	X
20	31.8	288, 352	C_15_H_17_NO_4_	-	-	276.1230	205, 190, 175, 162	Methoxy-methylenedioxy *trans*-cinnamoyl pyrrolidine	X	X
21	34.2	301	C_30_H_34_N_2_O_8_	-	-	551.2383	223, 152	Cyclobutene-2,4-*bis*-(1,3-benzodioxol-5-methoxy-6-yl)-1,3-dicarboxapyrrolidide	X	X
22	34.5	304	C_30_H_34_N_2_O_8_	-	-	551.2393	205, 190, 175, 162	Derivative dimeric of Methoxy-methylenedioxy *trans*-cinnamoyl pyrrolidine	X	X
23	35	277	C_18_H_21_NO_3_	-	-	300.1594	229, 187, 165, 157, 150	*N*-[7-(3′,4′)-methylenedioxyphenyl-2,4, heptadienoyl]- pyrrolidine	X	X
24	35.2	274	C_18_H_21_NO_3_	-	-	300.1587	246, 229, 187, 165, 150	*N*-[7-(3′,4′)-methylenedioxyphenyl-2,4, heptadienoyl]- pyrrolidine	X	X
25	35.3	280	C_18_H_21_NO_3_	-	-	300.1595	272, 258, 229, 201, 199, 187, 171, 161, 157, 150	*N*-[7-(3′,4′)-methylenedioxyphenyl-2,4, heptadienoyl]- pyrrolidine	X	X
26	35.8	294, 304	C_33_H_38_N_2_O_7_	-	-	575.2339	300, 276, 205, 190, 175, 162	Dimeric amide (methoxy-4′,5′-methylenedioxy-cinnamoyl pyrrolidide and *N*-[7-(3′,4′)-methylenedioxyphenyl-2,4, heptadienoyl]- pyrrolidine)	X	X
27	36.4	294, 304	C_33_H_38_N_2_O_7_	-	-	575.2741	391, 300, 276, 231, 205, 166,504, 391, 302, 276, 231, 205, 190, 165	Dimeric amide (methoxy-4′,5′-methylenedioxy-cinnamoyl pyrrolidide and *N*-[7-(3′,4′)-methylenedioxyphenyl-2,4, heptadienoyl]- pyrrolidine)		X
28	36.8	293, 304	C_33_H_38_N_2_O_7_	-	-	575.2746	504, 405, 300, 276, 205, 152	Dimeric amide		X
29	37.1	-	C_18_H_30_O_3_	293.2129	275, 249, 223, 205, 195, 167	-	-	Hydroxy-octadecatrienoic acid		X
30	32.4	300	C_20_H_25_NO_3_	-	-	328.1909	300, 299, 199, 187, 161	Methylenedioxyphenyl-*oxo*-nonadienyl pyrrolidine (brachyamide B)		X
31	37.9	300	C_20_H_27_NO_3_	-	-	330.2063	288, 259, 241, 208, 154	Methylenedioxyphenyl-*oxo* nonenyl pyrrolidine (tricholein)		X
32	37.9	310	C_34_H_38_N_2_O_6_	-	-	571.2799	500, 472, 458, 429, 401, 300, 272, 223, 201, 187, 175, 161, 152	Nigramide derivative		X
33	38.0	-	C_18_H_32_O_3_	295.2289	277, 195, 177	-	-	Hydroxy-octadecadienoic acid		X
34	39.6	307	C_22_H_29_NO_3_	-	-	356.2216	328, 285, 257, 234, 187, 161	Methylenedioxyphenyl-*oxo*- undecadienyl—pyrrolidine		X
35	39.8	299, 408	C_25_H_33_NO_4_	-	-	412.2473	276, 205, 190, 165	Unknown		X
36	40.1	307	C_22_H_31_NO_3_	-	-	358.2377	316, 287, 236, 177, 168, 161, 154	Methylenedioxyphenyl-*oxo*- undecenyl—pyrrolidine		X
37	40.4	-	C_24_H_31_NO_3_	-	-	382.2374	340, 260, 227, 213, 201, 197, 187, 173 161, 157, 154	Methylenedioxyphenyl-*oxo*- tridecatrienyl—pyrrolidine (Brachyamide A)		X
38	40.9	-	C_20_H_29_NO	-	-	310.3100	-	Unknown	X	
39	41.4	-	C_18_H_31_NO	-	-	278.2476	236, 168, 154, 149	Methylpropyl-tetradecatrienamide		
40	41.6	300	C_24_H_33_NO_3_	-	-	384.2525	313, 262, 175, 161	Methylenedioxyphenyl-*oxo*- tridecadienyl—pyrrolidine		X
41	42.1	307	C_24_H_35_NO_3_			386.2685	356, 344, 315, 264, 210, 182, 177, 154, 149	Methylenedioxyphenyl-*oxo*- tridecenyl—pyrrolidine		X
42	42.2	-	C_22_H_41_NO	-	-	336.3258	294, 280, 266, 252, 238, 224, 210, 196, 182, 168, 154	*N*-isobutyl octadecadienamide	X	
43	42.6	-	C_24_H_41_NO	-	-	360.3251	318, 294, 264,250, 247, 246, 236, 232, 224, 208, 204, 196, 194, 182, 166, 154	Unknown (amide)	X	
44	42.8	-	C_22_H_35_NO	-	-	330.2794	288, 259, 252, 246, 217, 180, 166,154	*Oxo*-octadecatetraenyl- pyrrolidine		X
45	43.8	-	C_20_H_35_NO	-	-	306.2792	278, 264, 250, 238, 223, 210, 208, 194, 180, 166, 154	*N*-Methylpropyl-hexadecatrienamide		X
46	44.3	-	C_22_H_37_NO	-	-	332.2946	304, 290, 261, 236, 194, 180, 166, 154	*Oxo*-octadecatrienyl- pyrrolidine		X
47	44.9	-	C_22_H_39_NO	-	-	334.3101	292, 278, 263, 250, 238, 224, 210, 196, 182, 168, 154	*Oxo*-octadecadienyl- pyrrolidine		X

NI: non-identified; RT: retention time; MF: molecular formula; all the MFs were determined from the errors and mSigma below 5 ppm and 30, respectively.

**Table 2 pharmaceuticals-16-01630-t002:** Chemical compounds identified from *Piper amalago* volatile oil by GC-MS.

Peak	RT (min)	RI^a^	RI^b^	Compound	Peak Area Rel. (%)
1	18.35	930	932	***α*-Pinene ***	**6.17**
2	19.14	942	946	Camphene *	1.1
3	20.91	969	974	*β*-Pinene *	3.77
4	21.78	982	988	*β*-Myrcene *	3.31
5	22.70	996	1002	*α*-Phellandrene	0.41
6	23.74	1011	1020	*p*-Cymene *	0.98
7	24.36	1020	1025	***β*-Phellandrene ***	**8.12**
8	24.44	1021	1024	Limonene *	1.58
9	28.88	1083	1095	Linalol	0.93
10	33.36	1146	1165	Camphol	0.33
11	33.75	1152	1183	Cryptone	0.59
12	34.28	1159	1174	Terpinen-4-ol *	0.24
13	35.02	1170	1186	*α*-Terpineol *	0.21
14	46.17	1334	-	NI	1.83
15	48.80	1374	1374	α-Copaene	0.31
16	49.57	1386	1389	*β*-Elemene	0.88
17	51.49	1416	1417	*β*-Caryophyllene *	3.97
18	52.64	1434	1439	Aromandendrene	0.75
19	53.44	1446	1452	*α*-Humulene *	0.69
20	53.80	1452	1457	*β*-Santalene	0.43
21	54.77	1467	1478	*γ*-Muurolene	2.05
22	55.10	1472	1484	Germacrene D	0.78
23	55.87	1484	1496	Valencene *	0.87
24	56.11	1487	1500	**Bicyclogermacrene**	**11.00**
25	56.30	1490	1500	*α*-Muurolene	0.85
26	56.87	1499	1505	*β*-Bisabolene	2.90
27	57.13	1503	1513	*γ*-Cadinene	1.46
28	57.36	1506	1521	Calamenene	0.33
29	57.74	1511	1522	δ-Cadinene	4.49
30	58.75	1526	1537	*α*-Cadinene	0.23
31	60.15	1545	1559	Germacrene B	3.47
32	60.80	1554	1567	Palustrol	0.35
33	60.99	1557	1577	**Spathulenol ***	**9.21**
34	61.44	1563	1582	Caryophyllene oxide *	1.15
35	61.73	1568	1590	Globulol *	1.04
36	62.25	1575	1592	Viridiflorol *	0.42
37	62.43	1577	1595	Cubeban-11-ol	0.65
38	63.10	1587	1602	Ledol	0.47
39	63.92	1598	1599	Widdrol	0.21
40	64.55	1606	1630	Muurola-4,10(14)-dien-1*β*-ol	0.22
41	64.83	1610	1618	1,10-di-*epi*-Cubenol	0.29
42	65.61	1620	1640	*epi-α*-Muurolol	3.14
43	65.81	1623	1644	*δ*-Cadinol	0.57
44	65.98	1625	1645	Cubenol	0.20
45	66.17	1627	1649	*β*-Eudesmol *	0.67
46	66.51	1632	1652	*α*-Cadinol	4.53
47	67.05	1639	1685	*ent*-Germacra-4(15),5,10(14)-trien-1*α*-ol	0.21
▪ Monoterpenes hydrocarbons					17.02
▪ Oxygenated monoterpenes					10.64
▪ Sesquiterpenes hydrocarbons					36.17
▪ Oxygenated sesquiterpenes					34.04
▪ Not identified					2.13

RT: retention time; NI: not identified. Identification based on the retention times (RTs) of genuine compounds in the DB-5MS column; MS, determined based on computer correspondence of the mass spectra with the spectra of the NIST library and comparison with literature data. Stereoisomers not identified; RI^a^, linear retention index experimentally obtained in this research; RI^b^, literature retention index (column DB-5) (Adams, 2017) [20]; * compound identification is confirmed with the reference standard. The components in bold are the major compounds.

**Table 3 pharmaceuticals-16-01630-t003:** Effects of acute oral administration of AE, HE, and VO obtained from *Piper amalago* leaves on urinary volume at 8 and 24 h.

Parameters	Control	AE	AE	HE	HE	VO	VO	HCTZ
		30 mg/kg	300 mg/kg	30 mg/kg	300 mg/kg	30 mg/kg	300 mg/kg	
DAY 3								
Urine volume (mL/100 g)	0.67 ± 0.02	1.17 ± 0.16 ^a^	1.04 ± 0.02 ^a^	1.29 ± 0.07 ^a^	1.18 ± 0.08 ^a^	0.80 ± 0.06	0.85 ± 0.02	1.20 ± 0.12 ^a^
DAY 7								
Urine volume (mL/100 g)	3.69 ± 0.30	7.77 ± 0.98 ^a^	5.06 ± 0.26	8.98 ± 0.86 ^a^	5.46 ± 0.62	2.80 ± 0.62	4.70 ± 0.34	3.73 ± 0.48

Values are expressed as mean ± standard error of the mean (*n* = 6). Statistical analysis was performed by one-way ANOVA followed by Dunnett’s test. ^a^ *p* ≤ 0.05 when compared with the control group. AE: aqueous extract; HE: hydroalcoholic extract; HCTZ: hydrochlorothiazide; VO: volatile oil.

**Table 4 pharmaceuticals-16-01630-t004:** Effect of acute oral administration of AE, HE, and VO obtained from *Piper amalago* leaves on electrolyte excretion at 8 and 24 h.

Parameters	Control	AE	AE	HE	HE	VO	VO	HCTZ
		30 mg/kg	300 mg/kg	30 mg/kg	300 mg/kg	30 mg/kg	300 mg/kg	
8 h								
Urea (mg/dL)	2141 ± 58.01	1366 ± 148.6 ^a^	1663 ± 118.9	1237 ± 11.81 ^a^	1434 ± 40.9 ^a^	2676 ± 259.6	2181 ± 148.1	1442 ±164.5 ^a^
Creatinine (mg/dL)	25.29 ± 0.18	16.96 ±1.18 ^a^	19.73 ±0.95	15.81 ± 0.26 ^a^	17.83 ± 0.78 ^a^	29.52 ± 4.14	27.67 ±1.47	16.06 ±1.92 ^a^
El_Ca_^++^	9.90 ± 0.27	6.84 ± 1.00	9.51 ± 1.04	9.27 ± 2.37	6.98 ± 0.66	7.09 ± 0.60	7.98 ± 0.59	7.51 ± 0.14
El_Mg_^++^	2.99 ± 0.66	2.39 ± 1.40	4.13 ± 1.23	3.68 ± 1.70	1.25 ± 0.36	7.87 ± 0.70 ^a^	6.07 ± 0.81	5.98 ± 0.89
El_Na_^+^	163.7 ± 12.55	175 ± 1.89	172.3 ± 6.58	154.5 ± 5.48	135 ± 11.82	202.3 ± 27.26	157.3 ± 3.14	169.3 ± 4.86
El_K_^+^	149.2 ± 2.06	116.1 ± 10.48	132.1 ± 8.15	106 ± 0.36 ^a^	111.8 ± 4.75 ^a^	153.9 ±13.89	142.3 ± 9.82	110 ± 7.99 ^a^
El_Cl_^−^	200.7 ± 10.93	201.3 ± 4.69	194.7 ± 6.52	171.7 ± 3.32	156.1 ± 11.79	233.3 ± 30.96	189.7 ± 1.99	194.2 ± 9.20
24 h								
Urea	2156 ± 112.3	1788 ± 176.8	2659 ± 293.8	1851 ± 222.5	2281 ± 236.7	3211 ± 251.5 ^a^	2445 ± 185.6	1952 ± 78.57
Creatinine	20.67 ±1.48	17.14 ±1.22	25.23 ± 2.10	14.54 ± 3.52	21.03 ±1.38	36.88 ±7.64 ^a^	25.29 ±1.65	17.06 ±0.70
El_Ca_^++^	4.07 ± 0.21	4.11 ± 0.24	7.18 ± 1.08 ^a^	5.95 ± 0.59	5.23 ± 0.61	4.83 ± 0.65	3.94 ± 0.43	3.59 ± 0.37
El_Mg_^++^	0.68 ± 0.06	0.58 ± 0.27	0.53 ± 0.11	1.41 ± 0.68	0.30 ± 0.06	1.08 ± 0.32	1.60 ± 0.67	0.51 ± 0.02
El_Na_^+^	94.67 ± 11.11	89.33 ± 6.11	117.3 ± 7.88	89 ± 7.95	104 ± 10.48	155.3 ± 28.05 ^a^	98.67 ± 0.91	84.33 ± 1.05
El_K_^+^	170.7 ± 7.88	142.5 ± 12.41	204.9 ± 19.94	144.9 ± 177.2	177.2 ± 18.68	224.1 ± 25.02	178.6 ± 2.30	148.4 ± 7.09
El_Cl_^−^	154.6 ± 11.25	124.1 ± 9.46	162.1 ± 9.86	124.7 ± 10.54	151.1 ± 14.31	217.9 ± 35.26 ^a^	151.6 ± 2.42	130.6 ± 3.06

Values are expressed as mean ± standard error of the mean (*n* = 6). Statistical analysis was performed by one-way ANOVA followed by Dunnett’s test. ^a^ *p* ≤ 0.05 when compared with the control group. Electrolyte values are presented as µEq/min/100g (body weight). AE: aqueous extract; HE: hydroalcoholic extract; HCTZ: hydrochlorothiazide; VO: volatile oil.

**Table 5 pharmaceuticals-16-01630-t005:** Effects of repeated 7-day oral dose administration of AE, HE, and VO obtained from *Piper amalago* leaves on 24-h urine volume on day 3 and day 7.

Parameters	Control	AE	AE	HE	HE	VO	VO	HCTZ
		30 mg/kg	300 mg/kg	30 mg/kg	300 mg/kg	30 mg/kg	300 mg/kg	
DAY 3								
Urine volume (mL/100 g)	6.16 ± 0.66	5.17 ± 0.22	4.08 ± 0.11	6.64 ± 0.39	3.32 ± 0.29	2.46 ± 0.59	4.14 ± 0.09	6.35 ± 1.05
DAY 7								
Urine volume (mL/100 g)	3.68 ± 0.68	9.87 ± 1.94 ^a^	8.26 ± 2.49	5.76 ± 0.76	4.43 ± 0.45	15.67 ± 1.46 ^a^	10.24 ± 2.48 ^a^	13.54 ± 2.05 ^a^

Values are expressed as mean ± standard error of the mean (*n* = 6). Statistical analysis was performed by one-way ANOVA followed by Dunnett’s test. ^a^ *p* ≤ 0.05 when compared with the control group. AE: aqueous extract; HE: hydroalcoholic extract; HCTZ: hydrochlorothiazide; VO: volatile oil.

**Table 6 pharmaceuticals-16-01630-t006:** Effect of repeated 7-day oral dose administration of AE, HE, and VO obtained from *Piper amalago* leaves on electrolyte excretion on day 3 and day 7 of treatment.

Parameters	Control	AE	AE	HE	HE	VO	VO	HCTZ
		30 mg/kg	300 mg/kg	30 mg/kg	300 mg/kg	30 mg/kg	300 mg/kg	
DAY 3								
Urea (mg/dL)	3370 ± 228.3	4001 ± 113.7	5020 ± 234.9	3777 ± 141.1	4566 ± 295.6	5724 ± 931.6 ^a^	4519 ± 110	2853 ±322.4
Creatinine (mg/dL)	8.62 ± 0.92	3.64 ± 0.33	9.25 ± 3.18	11.69 ± 5.68	28.16 ± 5.57	71.64 ± 31.06 ^a^	22.04 ± 3.23	8.65 ± 1.53
El_Ca_^++^	1.01 ± 0.17	3.3 ± 0.33	2.63 ± 0.17	2.28 ± 0.14	3.42 ± 0.36	2.2 ± 0.43	2.25 ± 0.11	1.12 ± 0.44
El_Mg_^++^	0.37 ± 0.04	0.26 ± 0.03	0.22 ± 0.02	0.17 ± 0.01	0.17 ± 0.02	0.41 ± 0.08	0.25 ± 0.05	1.03 ± 0.38
El_Na_^+^	71 ± 4.56	77 ± 6.22	105 ± 75.33	75.33 ± 5.85	99.33 ± 6.98	111.7 ± 18.34	91.33 ± 1.72	52.67 ± 4.93
El_K_^+^	203.9 ± 13.76	277.7 ± 8.53	340.6 ±9.57 ^a^	267.6 ± 11.98	320.9 ± 23.18 ^a^	332.7 ± 58.56 ^a^	262.7 ± 1.04	152 ± 18.1
El_Cl_^−^	122.5 ± 3.87	153. 5 ± 8.25	200.2 ±14.61	154.8 ± 8.40	208 ± 16.55	179.1 ± 25.76	160 ± 1.68	81.83 ± 9.53
DAY 7								
Urea (mg/dL)	848.5 ± 119.3	2544 ± 441.9 ^a^	208 ± 383.6	2325 ± 166.3	3000 ± 185.1 ^a^	773.1± 84.11	1747 ± 533.4	878. ± 36.38
Creatinine (mg/dL)	13.27 ± 1.44	3.82 ± 1.00	2.37 ± 0.31	2.45 ± 0.35	4.36 ± 0.82	13.23 ± 2.69	11.11 ± 1.71	11.62 ± 2.75
El_Ca++_	4.64 ± 1.34	2.02 ± 0.72	2.24 ± 0.28	2.12 ± 0.37	3.81 ± 0.71	0.6 ± 0.03 ^a^	2.05 ± 1.09	0.49 ± 0.03 ^a^
El_Mg++_	0.26 ± 0.04	0.3 ± 0.08	0.25 ± 0.07	0.26 ± 0.01	0.60 ± 0.17	0.19 ± 0.01	0.32 ± 0.08	0.23 ± 0.03
El_Na+_	14 ± 1.67	83.33 ± 7.69 ^a^	83.67 ± 12.27 ^a^	76 ± 6.70 ^a^	97.33 ± 11.78 ^a^	14.33 ± 1.47	42.67 ±16.27	22.67 ± 1.52
El_K+_	27.64 ± 1.76	248.9 ± 20.22 ^a^	258.9 ± 38.22 ^a^	279.4 ± 17.49 ^a^	317.1 ± 35.84 ^a^	29.92 ± 5.04	107.4 ±47.14	38.03 ± 2.99
El_Cl−_	20.47 ± 2.48	180.6 ± 14.37 ^a^	198 ± 29.38 ^a^	199.5 ± 11.02 ^a^	230.8 ± 30.25 ^a^	19 ± 2.26	71.7 ± 31.51	30.5 ± 3.10

Values are expressed as mean ± standard error of the mean (*n* = 6). Statistical analysis was performed by one-way ANOVA followed by Dunnett’s test. ^a^ *p* ≤ 0.05 when compared with the control group. Electrolyte values are presented as µEq/min/100g (body weight). AE: aqueous extract; HE: hydroalcoholic extract; HCTZ: hydrochlorothiazide; VO: volatile oil.

**Table 7 pharmaceuticals-16-01630-t007:** Effects of repeated 7-day dose treatments with AE, HE, and VO obtained from *Piper amalago* leaves on serum biochemical parameters.

Parameters	Control	AE	AE	HE	HE	VO	VO	HCTZ
		30 mg/kg	300 mg/kg	30 mg/kg	300 mg/kg	30 mg/kg	300 mg/kg	
Creatinine (mg/dL)	0.42 ± 0.07	0.35 ± 0.04	0.35 ± 0.04	0.32 ± 0.03	0.35 ± 0.005	0.40 ± 0.06	0.46 ± 0.04	0.36 ± 0.04
Urea	43.06 ± 2.04	37.17 ± 3.59	38.90 ± 2.52	40.46 ± 4.34	38.58 ± 1.07	41.82 ± 2.13	40.23 ± 1.51	42.46 ± 2.64
(mg/dL)								
Sodium	128.60 ± 4.20	132.20 ± 2.60	136.70 ± 2.29	136.40 ± 1.94	137.20 ± 1.53	130.30 ± 3.25	134.20 ± 1.40	135.20 ± 4.20
(mmol/L)								
Calcium	9.42 ± 0.73	9.41 ± 0.26	10 ± 0.18	9.77 ± 0.38	9.92 ± 0.46	9.28 ± 0.56	10 ± 0.24	9.39 ± 0.52
(mg/dL)								
Potassium (mmol/L)	4.16 ± 0.25	4.58 ± 0.40	4.50 ± 0.12	4.42 ± 0.35	4.73 ± 0.51	4.86 ± 0.33	4.20 ± 0.19	4.09 ± 0.28
Magnesium (mg/dL)	2.16 ± 0.15	2.46 ± 0.21	2.50 ± 0.08	2.42 ± 0.15	2.40 ± 0.19	2.34 ± 0.13	2.26 ± 0.10	2.43 ± 0.15

Values are expressed as mean ± standard error of the mean (*n* = 6). Statistical analysis was performed by one-way ANOVA followed by Dunnett’s test. AE: aqueous extract; HE: hydroalcoholic extract; HCTZ: hydrochlorothiazide; VO: volatile oil.

**Table 8 pharmaceuticals-16-01630-t008:** Effects of repeated 7-day dose treatments with AE, HE, and VO obtained from *Piper amalago* leaves on electrocardiographic parameters of Wistar rats.

Parameter	Control	AE	AE	HE	HE	VO	VO	HCTZ
		30 mg/kg	300 mg/kg	30 mg/kg	300 mg/kg	30 mg/kg	300 mg/kg	
Segment (ms)								
PR	42 ± 5.51	41.80 ± 3.05	50 ± 7.97	39.50 ± 2.60	42.20 ± 4.83	38 ± 2.51	35.40 ± 5.66	37 ± 3.14
QRS	43 ± 0.95	39.40 ± 1.50	38.83 ± 1.70	41.50 ± 1.19	42 ± 1.67	40.75 ± 1.49	42 ± 2.07	44.33 ± 2.11
QT	77± 6.23	74.80 ± 1.74	76.67 ± 4.93	70.25 ± 3.79	85.60 ± 4.23	72.75 ± 1.03	82.20 ± 2.76	81 ± 4.22
QTC	145.40 ± 12.02	145.80± 0.95	141.70 ± 10.37	134.80 ± 9.01	175.60 ± 11.46	135 ± 1.47	143.60 ± 4.91	146.20 ± 5.73
Wave (mv)								
P	0.046 ± 0.007	0.058± 0.014	0.036 ± 0.007	0.072 ± 0.021	0.054 ± 0.012	0.040 ± 0.007	0.058 ± 0.007	0.036 ± 0.008
Q	−0.022 ± 0.005	−0.010 ± 0.006	−0.00 ± 0.006	−0.010 ± 0.004	−0.032 ± 0.003	−0.032 ± 0.014	−0.016 ± 0.008	−0.016 ± 0.005
R	0.304 ± 0.027	0.286 ± 0.025	0.275 ± 0.016	0.272 ± 0.021	0.338 ± 0.046	0.320 ± 0.049	0.296 ± 0.040	0.236 ± 0.023
S	0.004 ± 0.025	0.012 ± 0.032	0.074 ± 0.029	0.065 ± 0.025	0.065 ± 0.025	0.0425 ± 0.023	0.044 ± 0.027	−0.040 ± 0.026

Values are expressed as mean ± standard error of the mean (*n* = 6). Statistical analysis was performed by one-way ANOVA followed by Dunnett’s test. AE: aqueous extract; HE: hydroalcoholic extract; HCTZ: hydrochlorothiazide; VO: volatile oil.

## Data Availability

Data is contained within the article.

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
