# Peer review of "The Cardiorenal Effects of Piper amalago Are Mediated by the Nitric Oxide/Cyclic Guanosine Monophosphate Pathway and the Voltage-Dependent Potassium Channels"

_pharmaceuticals, 2023, doi:10.3390/ph16111630_

Round 1

Reviewer 1 Report

Comments and Suggestions for Authors

Tha manuscript aimed to investigate the effects of P. amalago on renal and cardiovascular. The study ha a regional relevance and can contribute to therapeutiv strategies in the future. However, main converns must be considered in this evaluation:

- All text must be checked, regarding form and content. Some abreviattions are wrong (eg: sometimes AE and other times EA.

-Its not clear the experimental design. As suggetion, the authors must provide a figure showing the steps and timeline

- Why 7 days was chosen? 

- Why the authors used male and female? It is not clear when the authors used male and female. Is there any diference? 

- The authors must show the morphometric parameters of heart and kidney: HW/TL, KW/TL, HW/BW, KW/BW etc

- Why the authors did not show the hystology of cardiac tissue? 

- In the figure 2, the authors used male or female? Size bars must be included

- Line 180: there is no difference in SBP. The sentence must be rewritten. 

- The methodoly section must be rewritten. The authors must inlcude all the detail in all techniques. It is really hard to understand some experimets.

- ECG: why the authors did not showed the the curves?

- The study seems to be very descriptive. The authors must includ some data suggesting mechanisms. For example, the authors accompatined the animals in chronic time? the inflammation statuscould be evaluated.

- The conclusion can not be supported by the results presented. 

Comments on the Quality of English Language

The text must be submitted to english correction

Reviewer 2 Report

Comments and Suggestions for Authors

Dear Editor,

I carefully read the manuscript "The cardiorenal effects of Piper amalago are mediated by the nitric oxide/cyclic guanosine monophosphate pathway and voltage-dependent potassium channels".

My comments and suggestions for the authors are the following:

 - In the abstract, the authors should include some quantitative data. Sample size should also include in the abstract.

 - Lines 483-487: Statistical methods should be discussed in details. For example, the authors should specify which statistical tests they performed to verify the normal distribution of the variables.

 - How was the sample size (n of rats) calculated? More information should be included in the manuscript.

 - Tables: The abbreviations used should be specified at the bottom of the tables.

 - The limitations of the study should be further and more deeply discussed by the authors.

Comments on the Quality of English Language

Please, see my comments below.

Reviewer 3 Report

Comments and Suggestions for Authors

This article investigates the safety and renal and cardiovascular effects of preparations from Piper amalago leaves, including aqueous extract (AE), hydroalcoholic extract (HE), and volatile oil (VO). Chemical analysis identified flavones, organic acids, amino acids, and amides as major components. Acute toxicity studies in rats showed the preparations were safe at high doses. All preparations displayed diuretic effects when given acutely and for 7 days, increasing urine output and electrolyte excretion. The HE also showed vasodilatory effects in isolated mesenteric vascular beds, mediated by nitric oxide and potassium channel activation.

The study provides evidence that P. amalago leaves have diuretic activity and the HE has vascular effects, validating the ethnopharmacological use of this plant. While the mechanisms are not fully elucidated, the results suggest effects on kidney function and vascular tone underlie the biological activities. Further research is needed to identify the specific active compounds and characterize their pharmacological targets. Overall, this study scientifically demonstrates the therapeutic potential of P. amalago, especially for renal and cardiovascular conditions.

  • The doses used for the extracts (30 and 300 mg/kg) don't appear justified. More dose-response studies are needed.
  • Limited phytochemical analysis of the extracts - only major components identified. Extensive profiling is needed to identify the bioactive constituents.
  • Mechanism of diuretic effects not fully elucidated. Further studies on ion transporters, aquaporins, etc. needed.
  • Blood pressure effects only evaluated after 7 days. Longer treatment duration may be required to see significant changes.
  • Vasodilation experiments assessed acute effects only. Chronic studies needed.
  • Lack of positive controls in some experiments makes the results difficult to interpret.
  • No statistical analysis of blood pressure data.
  • Limited statistical analysis for vasodilation data. Multiple comparisons needed.
  • Lack of concentration-response curves for vasodilation makes EC50 values impossible to determine.

Round 2

Reviewer 1 Report

Comments and Suggestions for Authors

The authors answered all the points and the manuscript was considerally improved.

Comments on the Quality of English Language

Minor editing of English language required

Author Response

The manuscript was thoroughly reviewed in English language.

Reviewer 2 Report

Comments and Suggestions for Authors

Dear Editor,

I carefully read the revised version of the manuscript, that is significantly improved compared to the original version.

Author Response

Thank you for your comments. As no additional changes were suggested, we have conducted a new review in the English language.